# Characterization and Treatment Responsiveness of Genetically Engineered Ornithine Transcarbamylase-Deficient Pig

**DOI:** 10.3390/jcm10153226

**Published:** 2021-07-22

**Authors:** Shin Enosawa, Huai-Che Hsu, Yusuke Yanagi, Hitomi Matsunari, Ayuko Uchikura, Hiroshi Nagashima

**Affiliations:** 1Division for Advanced Medical Sciences, National Center for Child Health and Development, Tokyo 157-8535, Japan; ams4026@gmail.com; 2Center of Organ Transplantation, National Center for Child Health and Development, Tokyo 157-8535, Japan; yanagi-y@ncchd.go.jp; 3Department of Pediatric Surgery, Reproductive and Developmental Medicine, Kyushu University Graduate School of Medical Sciences, Fukuoka 812-8582, Japan; 4Meiji University International Institute for Bio-Resource Research, Kanagawa 214-8571, Japan; hitomim@meiji.ac.jp (H.M.); a_uchikura@meiji.ac.jp (A.U.); hnagas@meiji.ac.jp (H.N.)

**Keywords:** urea cycle disorder, ornithine transcarbamylase, pig, disease model

## Abstract

To develop novel medical technologies, pig disease models are invaluable especially in the final stages of translational research. Recently, we established a genetically engineered ornithine transcarbamylase-deficient (OTCD) pig strain. Here, we report its characterization and treatment responsiveness. OTCD pigs were obtained by mating an OTCD carrier female (*OTC*-X^c.186_190del^X^WT^) with a wild-type male. Due to the X-linked recessive mode of inheritance, the disease phenotype emerged only in males. Medication with nitrogen-scavenging agents was based on a clinical protocol. OTCD pigs were born smaller than their wild-type and carrier littermates, showing anemia and faltering. Biochemically, high levels of urinary orotic acid and loss of OTC activity were observed. The natural life course of OTCD pigs was characterized by a decrease in arterial percentage saturation of oxygen and body temperature, as well as an increase in blood ammonia levels; the pigs died in 24.0 ± 5.0 h (mean ± SD, *n* = 6). The established standard medication composed with nitrogen-scavenging agents and transfusion nearly doubled the survival time to 42.4 ± 13.7 h (*n* = 6). Our OTCD pig model appropriately mimicked the human pathology. Along with established protocols in handling and medication, this is a first step in developing a large animal disease model that is useful for translational research into novel medical technologies, such as cell transplantation and gene therapy, as well as in relation to urea cycle disorder.

## 1. Introduction

Most congenital metabolic disorders are rare orphan diseases, and the available therapies remain limited. Urea cycle disorder is a metabolic disorder caused by genetic mutation of the constitutive enzymes and it is characterized by hyperammonemia [1]. Recently, we developed a genetically engineered pig strain that lacks functional ornithine transcarbamylase (OTC), a constitutive enzyme in the urea cycle, with the aim of establishing a human disease model [2].

The urea cycle is composed of six enzymes and two transporters located in the mitochondria and cytoplasm of hepatic parenchymal cells and it is responsible for the endogenous production of arginine, ornithine, and citrulline, as well as the clearance of nitrogen end products that result from protein and other nitrogenous metabolic compound turnovers [3]. Genetic deficiencies in any of the enzymes or transporters cause urea cycle disorders and sometimes result in life-threatening symptoms including encephalopathy. Among the various patterns of urea cycle disorders, carbamoylphosphate synthetase I deficiency and OTC deficiency (OTCD) often develop severe clinical states characterized by hyperammonemia [3]. These enzymes are localized in the mitochondria, but the nature of the disease is completely different from mitochondrial liver diseases, whose etiology is characterized by the lack of a functional electron transport chain. Unlike mitochondrial liver diseases, urea cycle disorder’s fatal outcome, i.e., hyperammonemia, as well as other metabolic discrepancies are restrained during the gestation period by uteroplacental circulation; the diseases are often recognized after birth, unless fetal genetic diagnosis is performed, as is often for the case with at-risk babies. Thus, early appropriate treatments are of great importance for repressing the progress of encephalopathy and its resulting mental retardation. The mainstays of treatment are dietary protein restriction and administration of nitrogen-scavenging agents [3]. Although hemodialysis is the fastest method for lowering ammonia levels, neonatal patients should be paid the greatest attention when it comes to treatment, because the load on the cardiovascular system and the risks of infection by cannula insertion are much higher in these patients than in adults. Similarly, liver transplantation is not an indication for infantile patients, and it is usually performed after the stabilization of metabolic disorders at around 5 to 6 months of age [4].

The OTC gene is located on the X chromosome in humans and the infantile fulminant OTCD phenotype usually appears in males; however, the slow-onset, mildly progressive condition occurs in both females and males. The gene is also located on the X chromosome in pigs, and we succeeded in the establishment of fulminant OTCD model in male piglets, together with two other X-linked genetic disorders, namely severe combined immunodeficiency and Duchenne musculardystrophy [2]. In this study, we report the phenotypic characterization of OTCD pigs and discuss the foresight of pigs as large-scale experimental animal models.

Although nitrogen-scavenging agents are effective at decreasing blood ammonia, a complete cure can only currently be attained by liver transplantation. However, liver transplantation is invasive, especially in babies, and is limited due to donor shortages. Thus, less invasive, long-lasting treatments, such as cell transplantation or gene therapy, are desired, but so too is the development of novel drugs. Animal models of human disease are invaluable for developing novel medical technologies; here, we illustrate the characterization and treatment responsiveness of OTCD pigs as a core protocol for translational research in the future.

## 2. Materials and Methods

### 2.1. Animals

All animals used in this study were handled in accordance with the Japanese Guidelines for Animal Experiments of the Ministry of Health Labor and Welfare and the study was approved by the Institutional Animal Ethics Committee of the National Center for Child Health and Development (IRB number: A2000-001-C13) and Meiji University (IACUC11-0016, 12-0008, 14-0010, 15-0002, 15-0003). Genetically engineered OTCD pigs were established in a domestic Landrace–Large white–Durac strain by two-step breeding, as described previously [2]. Briefly, we established female pigs carrying heterozygous mutations in exon 2 of the OTC gene (c.186_190delTCTGA) using artificial reproductive technologies, including somatic cell cloning and blastocyst complementation. To obtain OTCD males, we performed planned mating of the carrier female and intact male at the estrous period. Most piglets were delivered naturally; however, some were treated with oxytocic or hysterotomy when the birth took too long after coming after amniorrhexis (Appendix A). OTCD piglets were kept in an incubator chamber for neonates (Caleo, Dräger, Lübeck, Germany) at a temperature of 38 °C and supplied up to 3 L/min of O_2_ in response to oxygen saturation monitored by an oxygen saturation monitor.

### 2.2. Genotyping

Genotyping was performed using restriction fragment length polymorphism analysis, as described previously [2]. Briefly, genomic DNA was extracted from tail biopsies of the offspring using a DNA extraction kit (DNeasy Blood and Tissue Kit, Qiagen, Hilden, Germany). The primers used were 5′- TCCAATCAGGCCTGTAGCTGC and 5′- GATTCCTTAGGTTCTGATTCAG. Nested PCR was then performed using PrimeSTAR HS DNA polymerase (Takara Bio) and the primers 5′- ACTGAGCAAGGTCCAGGATCG and 5′- TGCTATGCTCTTACACTCAGTC. The sequences of the amplicons containing the TALEN target region were determined using the sequencing primer 5′- TCACATCCTGAGTCTCCTCAAG. The PCR products were digested with the Hpy188I restriction enzyme (New England Biolabs, MA, USA), and the digested DNA fragments were analyzed by gel electrophoresis.

### 2.3. Medicines

The medicines used for OTCD pigs were as follows: sodium benzoate (Maruishi Pharmaceutical, Co., Ltd. Osaka, Japan), sodium phenylbutyrate (Buphenyl^®^, granules 94%, Horizon Therapeutics, Deerfield, IL, USA), L-arginine (Argi-U^®^ Injection, EA Pharma Co., Ltd., Tokyo, Japan), L-citrulline (KYOWA HAKKO BIO CO.,LTD., Tokyo, Japan), glucose (50%, Otsuka Pharmaceutical Co., Ltd., Tokyo, Japan), and lactate Ringer solution (SOLULACT^®^, Terumo Corporation, Tokyo, Japan).

### 2.4. Anesthesia, Analgesia, Surgical Procedures, and Blood Sampling

Anesthesia and analgesia for hysterotomy and catheter insertion were performed according to standard veterinary medicine protocols [5]. Briefly, pigs were sedated by intramuscular injection of 0.5 mg/kg mafoprazine mesylate (Mafropane^®^; DS Pharma Animal Health Co., Ltd., Osaka, Japan) followed by an intravenous injection of 1.5 mg/kg sodium thiopental (Ravonal^®^; NIPRO ES Pharma). Anesthesia was maintained by inhalation of 2.5% to 3.0% isoflurane (FUJIFILM Wako Pure Chemical Corporation, Osaka, Japan). As an analgesic, 0.3 mg/kg of butorphanol tartrate (Vetorphale^®^; Meiji Seika Pharma Co., Ltd., Japan) was administered intramuscularly before and after the operation. A single lumen 20-G catheter (1720-12-G, Covidien, Dublin, Ireland) was inserted into the jugular vein for drug infusion and blood sample correction. After the operation, the pigs were returned to the incubator chamber and left free. Infusion and blood sampling before cannulation were performed with the umbilical vein and tail vein, respectively, and in order to monitor the percentage saturation of oxygen from the tail or femoral artery.

### 2.5. Medication and Feeding

Standard care of OTCD piglets (Figure 1) was defined based on the protocol described in a professional review [1], with advice from medical staff in the Division of Endocrinology and Metabolism, National Center for Child Health and Development. OTCD piglets were infused intravenously with 13.5 mL/kg lactate Ringer solution containing 500 mg/kg sodium benzoate, 500 mg/kg L-arginine, and 0.56 g/kg glucose for 1 h, immediately after birth from the umbilical vein (indicated as “Loading” in Figure 1). After loading, the piglets were continuously infused with 120 mL/kg/day lactate Ringer solution containing 500 mg/kg/day sodium benzoate, 250 mg/kg/day L-arginine, and 13.4 g/kg glucose. The piglets were also orally administered 200 mg/kg sodium phenylbutyrate and 83 mg/kg L-citrulline in aqueous suspension (10 mL/kg) every 8 h. From 24 h after birth, piglets were fed artificial milk (10 mL/kg of 20% solution containing 0.5 g/kg protein and 0.04 g/kg lipid, Weanny Milk^®^, Nosan Corporation, Kanagawa, Japan) via a feeding tube (all the procedures are summarized in Figure 1). In the control group, the OTCD piglets received 60 mL/kg/day of 10% glucose intravenously. Pigs at the end stage were injected with 80 µg/kg of midazolam (Dormicum Injection, Maruishi Pharmaceutical, Co., Ltd. Osaka, Japan) intravenously or intramuscularly every 1 or 2 h for sedation, and survival times were determined by the cessation of a heartbeat as monitored by a cardiac electrogram.

### 2.6. Biochemical Analysis

Blood ammonia levels were determined using the bromocresol green method (PocketChem^TM^, PA-4140, Arkray, Inc., Kyoto, Japan, measurement range: 10 to 400 nitrogen-µg/dL) or, if the values exceed the detection range of the method, the indophenol method (Ammonia-Test-Wako kit [277-14401], FUJIFILM Wako Pure Chemical Corporation, Osaka, Japan). The values by the bromocresol green method were compensated by multiplying by 1.25, which is the ratio of molecular weights of nitrogen and ammonia. Arterial oxygen saturation was measured using a portable blood analyzer, iStat, and its cartridge, CG4+ (Abbott, Abbott Park, IL, USA). Urinary orotic acid levels and hepatic OTC activities were determined using a colorimetric assay [6]. Urinary orotic acid levels were standardized with urinary creatinine levels measured using a laboratory assay kit (LabAssay™ Creatinine [290-65901], FUJIFILM Wako Pure Chemical Corporation, Osaka, Japan). OTC activities were assayed with homogenates prepared from the liver of piglets of OTCD (*n* = 6) and wild-type male piglets (*n* = 5) just after birth (not autopsied liver), and they were expressed as ng citrulline production/min/mg protein. Protein content was determined using the Bradford method (Bio-Rad Protein Assay [5000001JA], Bio-Rad Laboratories, Inc. Hercules, CA, USA).

### 2.7. Statistical Analysis

Statistically significant results were determined using one-way ANOVA to eliminate the risk of type 1 error. Fisher’s protected least significant difference (PLSD) test (Figure 2, Figure 3 and Figure 4) or the Wilcoxon rank-sum test (Figure 5) by JMP11 (SAS Institute) were also used. Statistical significance was set at *p* value < 0.05.

## 3. Results

### 3.1. Feature of OTCD Piglets and Delivery Data

Body weight measurements (mean ± SD: number in parentheses denotes sample number) of male OTCD (*OTC*-X^c.186_190del^Y), male wild-type, female carrier (*OTC*-X^c.186_190del^X^wt^), and female wild-type piglets were 0.617 ± 0.148 kg (19), 1.147 ± 0.294 kg (23), 1.279 ± 0.310 (12), and 1.146 ± 0.256 (10), respectively (Figure 2a). OTCD piglets appeared small, anemic, and faltering (Figure 2b and Appendix A). The number of live OTCD males ranged from 0 to 2 (average ± SD: 1.2 ± 0.7, *n* = 18) at each delivery (Table 1 and details in Appendix A). The average number of total OTCD males, including dead pigs, was 2.6, which was 27% of the total number of piglets, nearly equal to the genetical theoretical value of 1/4. Notably, the gestation lengths for female carriers in normal delivery cases (111 ± 2.1, *n* = 11) were shorter than the standard period of 114 days [7]. A reduction in gestation period was seen irrespective of the artificial interventions, hysterotomy after the sign of delivery, or oxytocic treatment.

### 3.2. Characterization of OTCD Piglets

OTCD piglets born anemic with coldish skin were kept in an incubator chamber for neonates and recovery. However, arterial oxygen saturation and body temperature gradually decreased (Figure 3a,b) and the piglets died. As evidence of the disease, elevation of urinary orotic acid and loss of OTC were observed (Figure 3c,d). The activity of male OTCD piglets was not detectable in all six cases by the colorimetric assay.

### 3.3. Effect of Medication on Blood Ammonia Level and Survival

Changes in blood ammonia levels at birth and throughout the time course, with or without medication, were summarized in Figure 4. Male OTCD piglets showed high ammonia levels (318 ± 182 µg/dL, *n* = 10) at birth, while other genotypes, including male wild-type, female OTCD carrier, and female wild-type piglets, did not exceed the physiological range (83 ± 31 (*n* = 13), 78 ± 17 (*n* = 9), 97 ± 17 (*n* = 9) µg/dL, respectively). The levels in male OTCD piglets without medication increased rapidly with time (Figure 4b, dashed lines). In contrast, the piglets receiving medication showed a delay in ammonia increase (Figure 4b, solid lines), suggesting beneficial effects of the administration of nitrogen-scavenging agents. However, the ammonia level increased after a transient steady period. Conversely, piglets of other genotypes showed a decrease in blood ammonia after birth (Figure 4c), similar to that in human babies. Interestingly, the medication accelerated the decrease in wild-type males (Figure 4c, open square with solid line).

In response to the depression of increases in ammonia (Figure 4b), the OTCD piglets that received medication showed significant prolongation of survival times compared to the controls (Figure 5, with medication: 42.4 ± 13.7 h (mean ± SD, *n* = 6), without medication; 24.0 ± 5.0, *n* = 6).

## 4. Discussion

The urea cycle is a life-supporting biochemical pathway in the liver. Its role is the endogenous production of arginine, ornithine, and citrulline, as well as the clearance of nitrogen end products that result from protein and other nitrogenous metabolic com-pound turnovers. Urea cycle disorder poses a risk of hyperammonemia attack, which can likely lead to irreversible central nervous system damage. However, the direct relationship between hyperammonemia and encephalopathy is controversial [3]. The most common disorder of the cycle is caused by mutations in OTC [1]. As the OTC gene is linked to the X-chromosome, a severe fulminant OTCD syndrome emerges in males. To date, infantile patients are treated with protein-free diet, nitrogen-scavenging agents, and hemodialysis, depending on the severity of individual cases. Most patients are treated with these symptomatic therapies until liver transplantation becomes available [4]. However, although liver transplantation is an effective treatment, patients have to take immunosuppressive drugs on a lifelong basis, and risk of rejection and need for dietary restrictions exist long after transplantation. While cell transplantation and gene therapy are being examined as novel treatments [5,8,9,10], their safety and therapeutic efficacies have not yet been determined. From this point of view, human disease models in which clinical equivalent treatments are executable are preferable for the development of novel medical technologies.

We established a genetically engineered OTCD pig strain that aims to model human disease [2]. The location of the OTC gene is on the X chromosome in both pigs and humans. Due to the X-linked inheritance pattern, the pathogenic variant that our OTCD pig model harbored through the breeding scheme utilized in this study was only observed in male pigs. As shown in Table 1, the OTCD trait was found in 27% of the piglets born from an OTCD carrier mother. Unlike human fetuses, in which the metabolic abnormalities are compensated by uteroplacental circulation, the OTCD trait is likely to influence fetuses in pigs.

In humans, individuals with OTCD only develop the pathological condition after birth and the symptoms are sometimes overlooked for one to two days, resulting in delayed therapeutic care. More complicatedly, the intronic variant, which often goes unnoticed, can be a cause of late-onset OTCD [11]. Therefore, the OTCD trait should be diagnosed not only by conventional methodologies, but also by fetal genetic testing during pregnancy. On the other hand, fetal pigs likely suffer from OTCD-induced damages due to following factors observed in our study: (1) OTCD piglets were smaller than wild-type and carrier littermates; (2) OTCD piglets already showed hyperammonemia and high orotic acid urine at birth; (3) there were stillborn male OTCD fetuses that died shortly before delivery; (4) the gestation period was shorter than the average gestation period of normal pigs, suggesting that the mother delivered the diseased fetuses early and deliberately. The above nature of OTCD pigs may have complicated our experiment, and specific attention is necessary in the nursing protocol.

To promote the survival of OTCD piglets, we established a standardized protocol using nitrogen-scavenging agents, sodium benzoate, and sodium phenylbutyrate (Figure 1). The medication successfully decreased the blood ammonia levels of OTCD piglets and prolonged their survival (Figure 4b and Figure 5). Dietary protein restriction and nitrogen-scavenging agent treatment are therapeutic mainstays. OTCD piglets were fed a protein-free glucose solution on the first day and started milk feeding 24 h after birth. The protocol was set based on clinical management guidelines [1] and the advice of clinical staff. Clinical treatment changes flexibly in response to a patient’s condition. There are various options when it comes to drugs, doses, and medical procedures. However, in animal experiments, standardization is the most important aspect in order to achieve scientific reproducibility. As such, best clinical practice and experimental requirements were compromised. Based on the present protocol, we are able to evaluate the effectiveness of novel medical technologies.

To improve the usability of the present OTCD pig model, early therapeutic intervention is practical and effective. Although there were no systemic data, the OTCD piglets delivered before term by hysterotomy or oxytocic treatment showed less severe symptoms and better responsiveness to treatment. Another strategy is the establishment of genetically engineered pigs with truncated OTC proteins of low OTC activity. That being said, it might have been possible to retain a small amount of OTC activity by adjusting the position and degree of recombination. As we did not know how to regulate the position and degree of OTC activity, we performed a complete loss of activity in this case. Thus, the severe fulminant OTCD model was established. The well-known OTCD model animals, sparse-fur (spf/1) mice (ID156, Balb/c background spontaneous mutation mice) show 1/6 the level of OTC activity (wild-type mice: 317.8 ± 51.5 nmol/min/mg, OTCD mice: 53.7 ± 33.5 nmol/min/mg, [mean ± SD, *n* = 3], unpublished data) and survive for approximately two months (median, 47 days; range: 20–52) [5,10]. In our previous study, hepatocyte transplantation two days after birth significantly prolonged the survival time in OTCD mice [10]. A mild OTCD model will be more sensitive to detecting the effectiveness of treatments in pigs.

We also followed up the blood ammonia levels of the carrier female pigs to investigate the possibility of a late-onset decrease in OTC activity as shown in chimeric mice with humanized livers with OTCD patients [12]. A small number of carrier females showed a gradual increase in blood ammonia levels until about 1 month of age. However, the levels were normalized and no change was observed thereafter. Some carrier females subsequently calved successfully. Domestic pigs, which we used in this experiment, grow too quickly to conduct long-term observations. If the OTCD trait could be transferred to miniature pigs [13], it would allow for longer-term observation with many more pigs and may allow for the study of the delayed-onset model of OTCD.

Large animal experiments have contributed greatly to medical innovation. The experimental system is established by animal resources and standardized protocols. The present system contributes to the development of innovative treatments for OTCD.

## 5. Conclusions

Our OTCD pig model appropriately mimicked the human pathology of the disease. Together with established protocols in handling and medication, it is a first step in developing a large animal disease model that is useful for novel medical technologies, such as cell transplantation and gene therapy, in relation to urea cycle disorder, especially in the late phase of translational research.

## Figures and Tables

**Figure 1 jcm-10-03226-f001:**
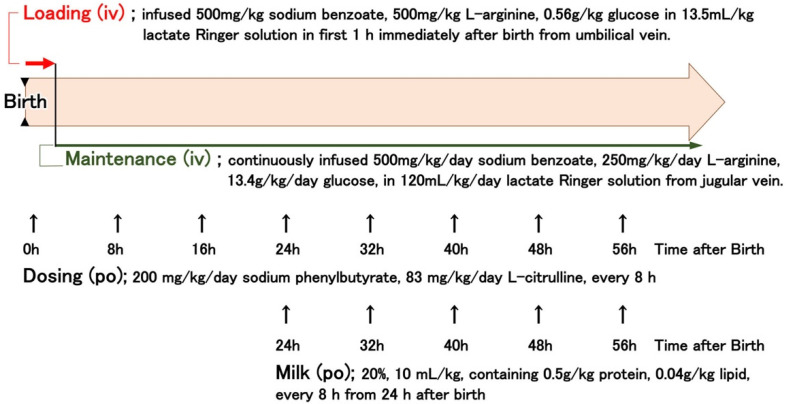
Treatment protocol of OTCD piglets. Control OTCD pigs received 60 mL/kg/day of 10% glucose intravenously. iv: intravenously, po: perorally.

**Figure 2 jcm-10-03226-f002:**
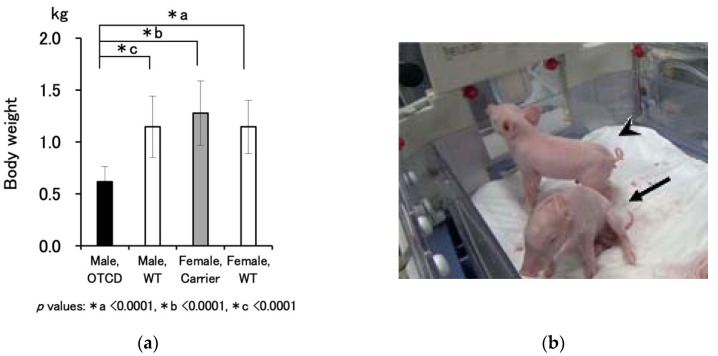
Comparison of the body size of piglets of each genotype: (**a**) body weight, kg (mean ± SD) of live piglets: male OTCD (*n* = 19), male wild-type (WT) (*n* = 23), female OTCD carrier (Carrier) (*n* = 12), and female WT (*n* = 10). Details of pregnancy and delivery outcomes are shown in Appendix A. The *p* value of the one-way ANOVA is <0.0001 and the asterisks denote statistical significances by Fisher’s PLSD test. (**b**) Typical appearance of OTCD (arrowhead, pig ID: K80-10) and WT (arrow, pig ID: K80-09) piglets 4 h after birth (littermates of mother pig K80 in Appendix A); also see Appendix A.

**Figure 3 jcm-10-03226-f003:**
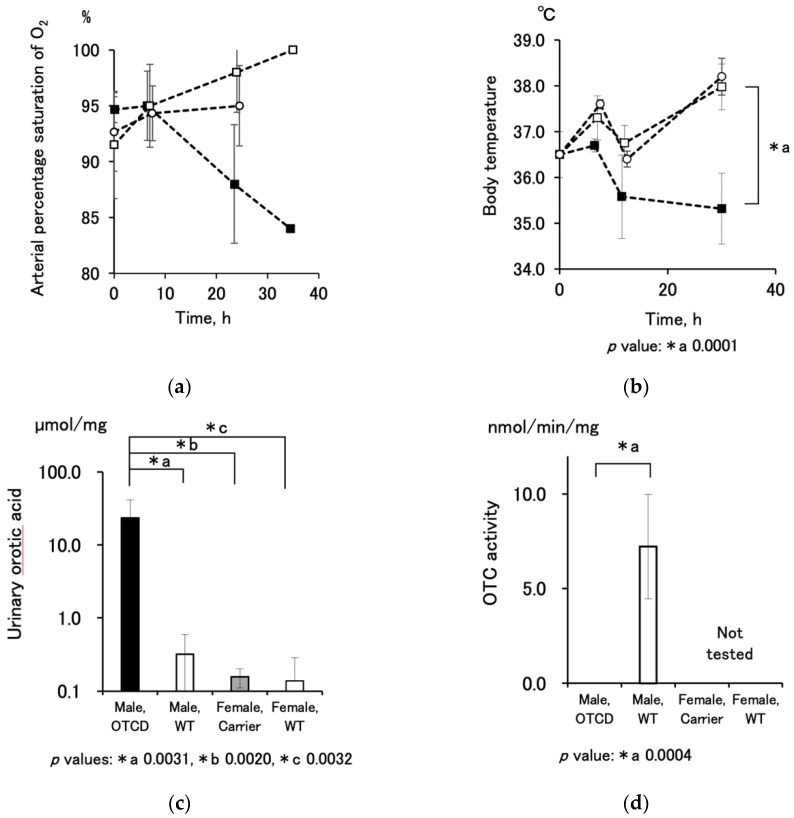
Characterization of OTCD piglets: (**a**) time course of arterial oxygen saturation, % (mean ± SD, *n* = 3–5 (points at 37 h without SD bar: *n* = 1)). Male OTCD, closed square; male WT, open square; female WT, open circle. (**b**) Time course of body temperature, °C (mean ± SD, *n* = 3–5). Male OTCD, closed square; male WT, open square; female WT, open circle. The *p* value of the one-way ANOVA was 0.0001 and the asterisk denotes statistical significance by Fisher’s PLSD test. (**c**) Comparison of orotic acid contents, µg/mg in urine taken within 2 h after birth (mean ± SD, *n* = 5, 4, 5, and 4, respectively). The values were expressed as µmol orotic acid/mg creatinine to counterbalance urinary density. The *p* value of the one-way ANOVA was 0.0045 and the asterisks denote statistical significance by Fisher’s PLSD test. (**d**) Comparison of hepatic OTC activities, nmol/min/mg (mean ± SD, *n* = 6 and 5, respectively). The values were expressed as nmol citrulline production/min/mg protein. The *p* value of the one-way ANOVA was 0.0004 and the asterisk denotes statistical significance by Fisher’s PLSD test and the *p* value 0.0004.

**Figure 4 jcm-10-03226-f004:**
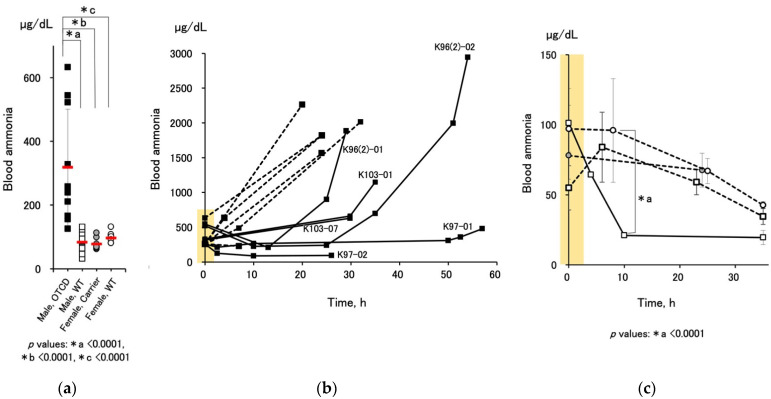
Comparison of blood ammonia levels, µg/dL of various condition: (**a**) blood ammonia levels in male OTCD (*n* = 10), male WT (*n* = 13), female OTCD carrier (*n* = 9), and female WT (*n* = 9) within 2 h after birth (values in the area highlighted with yellow in Figure 4b,c.). Red horizontal bars indicate average ± SD of each group. The *p* value of the one-way ANOVA was <0.0001 and the asterisks denote statistical significance by Fisher’s PLSD test. (**b**) Time course change in blood ammonia in individual male OTCD piglets with (solid line) or without (dashed line) medication. In the medication group (solid lines), corresponding pig IDs were indicated (see also Appendix A). K97-02 missed the last sampling before death due to technical difficulties. (**c**) Time course of blood ammonia levels in piglets with normal phenotypes (mean ± SD, *n* = 3–9). Male WT, open square; female carrier, gray circle; female WT, open circle. Dashed line, without medication; solid line, with medication. The *p* value of the one-way ANOVA was <0.0001 and the asterisk denotes statistical significance by Fisher’s PLSD test.

**Figure 5 jcm-10-03226-f005:**
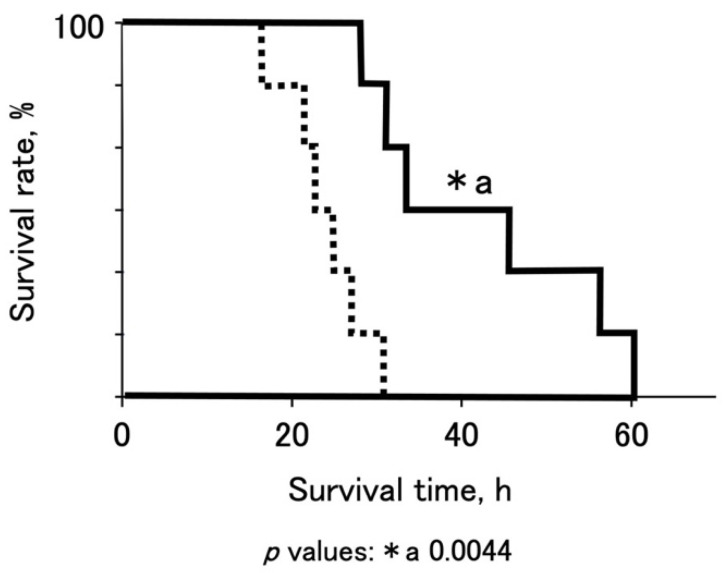
Prolongation of survival time, hof OTCD pigs by medication. Dashed line indicates survival times of OTCD pigs without medication and solid line indicates those of OTCD pigs with medication. The asterisk denotes statistical significance by Wilcoxon rank-sum test.

**Table 1 jcm-10-03226-t001:** Summary of pregnancy and delivery outcomes for ornithine transcarbamylase-deficient (OTCD) carrier female pigs ^1^.

	Average	SD
Gestation length, day	111.0	2.2
Divergence 2	−3.1	2.2
Mode of delivery 3	N: 11, H: 5, O: 2, Total *n* = 18	
Total No. of piglets	9.6	2.7
No. of live piglets	8.5	2.1
Live, male, OTCD 4	1.2	0.7
Live, male, WT 5	2.2	1.4
Live, female, carrier	2.6	1.5
Live, female, WT	2.3	1.6
No. of dead piglets	1.1	1.5
Dead, male, OTCD	1.1	1.5
Dead, male, WT	0.1	0.2
Dead, female, carrier	0.0	0.0
Dead, female, WT	0.0	0.0

^1^ Results were obtained from a total of 18 deliveries. Details of individual deliveries are shown in Appendix A. ^2^ Divergence from standard gestation period (114 days [7]). ^3^ N: normal delivery, H: hysterotomy after the sign of delivery, O: with oxytocic treatment. ^4^ OTCD: ornithine transcarbamylase deficient. ^5^ WT: wild-type.

## Data Availability

The data that support the findings of this study are available from the corresponding author, S.E., upon reasonable request.

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
