# Peer review of "Characterization and Treatment Responsiveness of Genetically Engineered Ornithine Transcarbamylase-Deficient Pig"

_jcm, 2021, doi:10.3390/jcm10153226_

Round 1

Reviewer 1 Report

Thank you for making the changes to the statistics that I commented on in my last review. I think this is acceptable. 

Author Response

Reply to reviewer 1
I appreciate very much for your informative comments up to this point.

Comment;
English language and style are fine/minor spell check required.

Answer; According to your last suggestion, I checked the manuscript carefully and completed MDPI’s English edit service.

Reviewer 2 Report

The manuscript has improved and now reads better than in its initial version. However, there remain some issues that authors should consider/correct/address, before this reviewer can recommend acceptance.

  1. Conceptual: authors expect their model to "be useful to develop ...". Since animals died within 31 h after birth, how shoud this model be useful for the given purpose? Any intervention would need to be initiated and already working on day 1.
  2. There is no clear information on the genetic background of the OTC pig (other than in the form of a citation of previous work). Please clearly state in abstract , methods and results that an OTC knock-out was done (and mention details).
  3. Add average of survival time in abstract instead of giving the maximum survival time (31 h).
  4. Do authors also have amino acid profiles from the pigs? This would be nice to see.
  5. Figure 1: consider indicating the timescale with increasing hours (8h, 16h, 24h, etc) instead of only indicating the 8h intervals
  6. Exchange "infants" by "piglets" throughout the paper
  7. Consider presenting results (from 3.1) in a table instead of a text
  8. Figure 2: add levels of significance to panels 2a), 3c) and 3d) and 4a) instead of listing them in legend
  9. Indicate when when exactly liver biopsies were done
  10. The data presentation in Figure 3d is misleading. If measured activities are considered errors, they should not be shown as column in 3d).
  11. Fix y-axes legends in Fig. 3c and 3d and 4a and 4b.
  12. The discussion is more or less a repetition of introduction and of results. Please rewrite to truely discuss your findings.
  13. There are still numerous spelling errors, eg. "meles" for males, "oxigen" for oxygen. Please thoroughly correct.
  14. There are still numerous grammatical errors, eg. unnecessary blanks, blanks missing, brackets missing, needless dots, etc. Please correct. 

Author Response

Reply to reviewer 2
Thank you very much for your careful reading and helpful comments. We did our best to address your comments.

The manuscript has improved and now reads better than in its initial version. However, there remain some issues that authors should consider/correct/address, before this reviewer can recommend acceptance.
Conceptual: authors expect their model to "be useful to develop ...". Since animals died within 31 h after birth, how should this model be useful for the given purpose? Any intervention would need to be initiated and already working on day 1.

Answer; According to your suggestion, I changed the phrase to be more truthful from "be useful to develop ..." to " be a first step to develop a large animal disease model useful for translational research of ..." (Lines 28-31 in Abstract, Lines 339-342 in Conclusion). In addition, I added the limitation of the present model and future solutions in Discussion.

1. There is no clear information on the genetic background of the OTCD pig (other than in the form of a citation of previous work). Please clearly state in abstract, methods and results that an OTC knock-out was done (and mention details).
Answer; We added the methodology and genetic background of OTCD pigs as follows. In Abstract, we added the words "OTC-Xc.186_190delXWT" (Lines 18-19). In Materials and Methods, we added the sentences, "we established female pigs carrying heterozygous mutation in exon 2 of the OTC gene (c.186_190delTCTGA) using artificial reproductive technologies including somatic cell cloning and blastocyst complementation.” (Lines 87-90). In section 3.1, Results, we added supplementary statements, (OTC-Xc.186_190delY) for OTCD males and (OTC-Xc.186_190delXwt) for female carrier (Line 178).

2. Add average of survival time in abstract instead of giving the maximum survival time (31 h).
Answer; I changed the data expression of survival time to “24.0 ± 5.0 h (mean ± SD, n=6)”. I also added the survival time of the medication group, 42.4 ± 13.7 h (n=6) to make Abstract more informative. (Lines 25 and 27)

3. Do authors also have amino acid profiles from the pigs? This would be nice to see.
Answer; I am very sorry but we do not have data on amino acid profiles.

4. Figure 1: consider indicating the timescale with increasing hours (8h, 16h, 24h, etc) instead of only indicating the 8h intervals
Answer; According to your suggestion, I changed the expression of timescales and replaced the Figure 1 by revised Figure 1. This change definitely made the figure more practical for researchers.

5. Exchange "infants" by "piglets" throughout the paper.
Answer; I am very sorry I had not completed the change despite being warned before. I carefully searched the word again and changed to "piglets".

6. Consider presenting results (from 3.1) in a table instead of a text.
Answer; According to your suggestion, I added Table 1 which summarized averages and SDs of supplementary table 1. Instead, I omitted the averages and SDs in supplementary table 1. Also, I transferred the sentence of footnote No.6 in supplementary table 1 to the end of section 3.1.

7. Figures: add levels of significance to panels 2a), 3c) and 3d) and 4a) instead of listing them in legend.
Answer; According to your comments, I showed p value(s) in the figures.

8. Indicate when exactly liver biopsies were done.
Answer; I am sorry for my carelessness. I added the description “OTCD (n=6) and wild type male piglets (n=5) just after birth” in Materials and Methods (Lines 165-166).

9. The data presentation in Figure 3d is misleading. If measured activities are considered errors, they should not be shown as column in 3d).
Answer; Thank you for your logical comment. I omitted the column of OTC activity of OTCD male piglets. Also, the relevant sentences in the text (Lines 207-208).

10. Fix y-axes legends in Fig. 3c and 3d and 4a and 4b.
Answer; I am sorry for my carelessness. I added the description to clarify the objectives to be shown in the figure legends (Lines 213 and 216).

11. The discussion is more or less a repetition of introduction and of results. Please rewrite to truly discuss your findings.
Answer; Thank you for your suggestion. I added the description of findings that I think will be useful to readers as follows: 1) potential usefulness of a porcine model of low OTC activity (Lines 313-317), 2) potentiality of late-onset OTCD pig models using miniature pigs (Line 324-332).

12. There are still numerous spelling errors, eg. "meles" for males, "oxigen" for oxygen. Please thoroughly correct.
Answer; I am very sorry for my carelessness. I checked the manuscript carefully and completed MDPI’s English edit service.

13. There are still numerous grammatical errors, eg. unnecessary blanks, blanks missing, brackets missing, needless dots, etc. Please correct.
Answer; I am very sorry for my carelessness. I checked the manuscript carefully and completed MDPI’s English edit service.

Round 2

Reviewer 2 Report

The revised paper has much improved and most issues are adressed and solved. I want to thank the authors for this. Below, please find some remaining minor issues that should be adressed in addition:

  1. Please re-phrase the sentence on page 3, line 58 ff, that starts "Although hemodialysis is the fastest ..." as the meaning of the second part of this sentence is unclear 
  2. In Fig. 1: "contentiously" may not be the correct word, consider instead "continuously" or else
  3. Page 3, lines 180/181: it is Figure 2 and not Figure 1a or 1b that is described here; please correct
  4. Table 1: please correct "infants" to "piglets" in this table
  5. Legend to Fig. 3: the comment in line 211 "[Points at 37 h without SD bar ...]" should be moved in the legend to Fig. 3a instead of 3b
  6. Legend to Fig. 3: line 212, add "WT" to "female"
  7. Comment for Fig. 2, 3, and 4: correct all the y-axes by including al units behind the description, for instance "kg" behind "body weight" in Fig. 2, etc 
  8. "Genetically" is correct (in title), no Need to Change to "geneticaly"

Author Response

Reply to Reviewer 2
   Thank you very much for your careful review. I did my best to address your comments.

Comments by Reviewer 2
The revised paper has much improved and most issues are addressed and solved. I want to thank the authors for this. Below, please find some remaining minor issues that should be addressed in addition:
1. Please re-phrase the sentence on page 3, line 58 ff, that starts "Although hemodialysis is the fastest ..." as the meaning of the second part of this sentence is unclear.
Answer: 
Thank you for your helpful comment. I added the description “because the load on cardiovascular system and the risks of infection by cannula insertion is much higher than in adults” to the sentence.

2. In Fig. 1: "contentiously" may not be the correct word, consider instead "continuously" or else
Answer:
I am sorry for my carelessness. I corrected "contentiously" to "continuously".

3. Page 3, lines 180/181: it is Figure 2 and not Figure 1a or 1b that is described here; please correct
Answer:
I am sorry for my carelessness. I corrected "Figure 1a or 1b" to "Figure 2".

4. Table 1: please correct "infants" to "piglets" in this table
Answer:
I am sorry to miss the words that you had pointed previously. I corrected "infants" to "piglets" in Table 1 and Supplementary table 1.

5. Legend to Fig. 3: the comment in line 211 "[Points at 37 h without SD bar ...]" should be moved in the legend to Fig. 3a instead of 3b
Answer:
I am sorry to misplace the phrase. I moved the phrase to an appropriate place in Fig.3a.

6. Legend to Fig. 3: line 212, add "WT" to "female"
Answer:
I am sorry for my carelessness. I added "WT" to "female".

7. Comment for Fig. 2, 3, and 4: correct all the y-axes by including al units behind the description, for instance "kg" behind "body weight" in Fig. 2, etc 
Answer:
I am sorry not to comprehend what you mean in previous review. I added the units in appropriate places in the figure legends.

8. ”Genetically" is correct (in title), no Need to Change to "geneticaly"
Answer: 
I am sorry for misspelling. I added “l” to the word.

This manuscript is a resubmission of an earlier submission. The following is a list of the peer review reports and author responses from that submission.

Round 1

Reviewer 1 Report

The paper by Enosawa and colleagues report on their experience in early treatment of OTC deficient pigs. They had performed (and published before) knock-out of the porcine OTC gene and now observed and treated piglets born with male and female OTC disease. The model they present is potentially important, since it is a large animal of a severe genetic hepatic disease. In such model, novel treatment approaches could be tested, thereby being an important translational tool.

This reviewer has 2 main issues to criticize:

  1. Since authors present a knock-out of OTC, animals survive for only 2 days (untreated) or 4 days (if treated with medication and diet). This obvious severity make any early postnatal intervention very challenging. For instance, cell transplantation or gene therapy (these are the modalities the authors mention themselves) would need to be initiated immediately after birth and would need to be effective without any delay. Both these requirements seem very challenging if not impossible to achieve. How realistic is it that this OTC-KO model will be as useful as authors described it? An obvious alternative would be to generate a less severe affected large animal model. Authors should critically reflect on this point and discuss this at least to some extent.
  2. The present paper is difficult to read because of the poor English. Some sentences or phrases are impossible to understand, eg. “hereditary pattern has been realized”, “there were not a few stillborn male OTCD fetuses that appeared to die shortly before delivery”. This paper needs to be thoroughly revised by an English native in close collaboration with authors to ensure the correct meaning.

In addition to these major points, there are some minor issues:

  1. Correct “infants” to “piglets” throughout the paper
  2. Check the detection limit of the Amicheck Meter; is it really 400 ug/dl?
  3. When discussing the gestation length, indicate the normal duration of your pigs
  4. In Figure 2, were all male and female OTCD pigs alive when born? In general, clearly indicate how many of the piglets were stillborn or alive in the affected pregnancies.
  5. Delete last sentence in legend to Figure 2 as it is redundant
  6. Add visual legend to Figure 3; it is cumbersome to identify the curves from the legend (closed square, open square etc)
  7. Male OTCD pigs still have some relevant OTC activity despite the OTC KO; how do authors explain this? How can this be reconciled with the severe phenotype?
  8. Figure 3a: are females shown on this graph WT or carriers?
  9. Figure 4: legend says “highlight of yellow area” but there is no such area in Fig. 4a; thus correct to 4b and 4c.

Author Response

Response to Reviewer 1

Dear Reviewer 1

Thank you very much for evaluating potential importance of our research and we sincerely appreciate your invaluable and warm comments. First of all, I have to apologize for submitting the manuscript without an adequate edit by a native speaker in English. I hereby submit a revised manuscript with track changes and response to the comments in point-by-point fashion.

Reviewer 1

The paper by Enosawa and colleagues report on their experience in early treatment of OTC deficient pigs. They had performed (and published before) knock-out of the porcine OTC gene and now observed and treated piglets born with male and female OTC disease. The model they present is potentially important, since it is a large animal of a severe genetic hepatic disease. In such model, novel treatment approaches could be tested, thereby being an important translational tool.

This reviewer has 2 main issues to criticize:

  1. Since authors present a knock-out of OTC, animals survive for only 2 days (untreated) or 4 days (if treated with medication and diet). This obvious severity makes any early postnatal intervention very challenging. For instance, cell transplantation or gene therapy (these are the modalities the authors mention themselves) would need to be initiated immediately after birth and would need to be effective without any delay. Both these requirements seem very challenging if not impossible to achieve. How realistic is it that this OTC-KO model will be as useful as authors described it? An obvious alternative would be to generate a less severe affected large animal model. Authors should critically reflect on this point and discuss this at least to some extent.

Response: Thank you for your on-target comment. We added our concept of experimental use in translational research and future provision in Discussion (line 297 - 310) .

  1. The present paper is difficult to read because of the poor English. Some sentences or phrases are impossible to understand, eg. “hereditary pattern has been realized”, “there were not a few stillborn male OTCD fetuses that appeared to die shortly before delivery”. This paper needs to be thoroughly revised by an English native in close collaboration with authors to ensure the correct meaning.

Response: Again, I apologize to submit without an adequate edit. We improved the sentence carefully and asked an edit to a professional English editor who knows our research well  ( https://www.editage.com ).

In addition to these major points, there are some minor issues:

  1. Correct “infants” to “piglets” throughout the paper.

Response: I changed the word “infants” to “piglets” throughout the paper.

  1. Check the detection limit of the Amicheck Meter; is it really 400 µg/dl?

Response: Thank you very much for noting this. The writing was misleading and I changed ‘>400 µg/dl’ to ‘measurement range; 10 to 400 nitrogen-µg/dl’. In addition, I changed following two points according to the manufacturer’s instruction  ( http://www.arkray.co.jp/english/upload/docs/pa_4140.pdf ) ; 1) The name Amicheck was changed to the present name, PocketChemTM, PA-4140unit, 2) Because the measurement unit of PocketChem is nitrogen-µg/dl, I added the description about compensation of the unit, ‘The values by the bromocresol green method were compensated by multiplying 1.25, the ratio of molecular weights of nitogen and annmonia.’

  1. When discussing the gestation length, indicate the normal duration of your pigs.

Response: I added reference 6 for the normal gestation length (114 days) 

  1. In Figure 2, were all male and female OTCD pigs alive when born? In general, clearly indicate how many of the piglets were stillborn or alive in the affected pregnancies.

Response: I am very sorry for the incomplete description. The results of Figure 2a were from all live piglets. I added the word ‘live’ and ‘Details of pregnancy and delivery outcomes were shown in Supplementary table 1’ in the legend. In addition, I added the data of ‘Total No.’ in Supplementary table 1 to view the number of live and stillborn piglets.

  1. Delete last sentence in legend to Figure 2 as it is redundant.

Response: The sentence ‘OTCD pig was small-sized, anemic, and faltering with coldish skin’ was deleted.

  1. Add visual legend to Figure 3; it is cumbersome to identify the curves from the legend (closed square, open square etc)

Response: I am very sorry but I could not insert the graphics in the MS Word text. I would be grateful if the publisher helps me this on the proof.

  1. Male OTCD pigs still have some relevant OTC activity despite the OTC KO; how do authors explain this? How can this be reconciled with the severe phenotype?

Response: The activities of male OTCD piglets in Figure 3d were considered as measurement error and I added the sentence ‘The activities of male OTCD piglets in Figure 3d were measurement errors in the colorimetric assay and considered to be zero.’ in the text.

  1. Figure 3a: are females shown on this graph WT or carriers?

Response: I am sorry for the incomplete description. These data were from wild type females. I added ‘WT’ in the Figure legend.

  1. Figure 4: legend says “highlight of yellow area” but there is no such area in Fig. 4a; thus correct to 4b and 4c.

Response: I am sorry for the mistake. I changed to 4b and 4c.

Reviewer 2 Report

This paper characterizes an OTC-deficient pig model for which the development was described in Reference #2. 

1) Paper provides phenotype characterization of the OTC-deficient pig model. The experiments described do not provide new information on the pathophysiology of OTC deficiency. Please replace 'pathophysiology' with 'characterization' throughout the manuscript. This includes changing the titled to "Characterization and Treatment Responsiveness of an Ornithine Transcarbamylase-Deficient Pig Model". 

2) Authors frequently refer to the OTC-deficient piglet model as OTC infants. This makes it sounds like a human study, which it is not. Throughout the manuscript, please use the terms OTC-deficient pig or OTC-deficient piglet. 

3) Please report genotyping methods.

4) Majority of groups that characterize new model organisms generated for urea cycle disorders include plasma amino acid data. Plasma amino acid data for the OTC-deficient piglets compared to WT littermates are missing and should be included. 

5) Statistical methods:

A) The one-way ANOVA model was primarily used with the Fisher's LSD to determine the nature of differences observed. Please include the ANOVA model p-values throughout the manuscript. They are missing.

B) The repeated measures ANOVA is required for the analyses shown in Figure 3A, 3B, and 4C. 

C) Was a t-test used for 3D? If yes, please indicate this in the figure legend and add to the methods.

6) Important background information on OTC deficiency was either missing or incorrect in the introduction and the discussion. I encourage the authors to carefully review more current literature to incorporate what the urea cycle is, inheritance patterns of OTCD, clinical presentations, long-term consequences, current treatments, in addition to other model organisms that have been reported for OTC deficiency. I made specific notes on this.

7) I made specific comments throughout a .pdf version of the manuscript. Please review and revise per the comments. 

8) Supplementary Table 1 - Revise title to "Pregnancy and Delivery Outcomes for OTC Carrier Female Pigs". Edit the font and sizes of the table content to be more consistent. 

Author Response

Response to Reviewer 2

Dear Reviewer 2

Thank you very much for evaluating our work and invaluable warm comments as well as introduction of helpful literature and careful edits in the manuscript. I must apologize for submitting the manuscript without adequate edit. I hereby submit a revised manuscript with track changes and response to the comments in point-by-point fashion.

This paper characterizes an OTC-deficient pig model for which the development was described in Reference #2.

1) Paper provides phenotype characterization of the OTC deficient pig model. The experiments described do not provide new information on the pathophysiology of OTC deficiency. Please replace 'pathophysiology' with 'characterization' throughout the manuscript. This includes changing the titled to "Characterization and Treatment Responsiveness of an Ornithine Transcarbamylase-Deficient Pig Model".

Response: Thank you very much for a thoughtful advice. I changed the word 'pathophysiology' to 'characterization' throughout the manuscript.

2) Authors frequently refer to the OTC-deficient piglet model as OTC infants. This makes it sounds like a human study, which it is not. Throughout the manuscript, please use the terms OTC-deficient pig or OTC-deficient piglet.

Response: Thank you very much for a thoughtful advice. I changed the word 'infants' to 'piglets' throughout the manuscript.

3) Please report genotyping methods.

Response: I am sorry not to describe the methods. I added the description as 2.2 Genotyping in Materials and Methods section.

4) Majority of groups that characterize new model organisms generated for urea cycle disorders include plasma amino acid data. Plasma amino acid data for the OTC-deficient piglets compared to WT littermates are missing and should be included.

Response: I am very sorry but we did not analyze plasma amino acids.

5) Statistical methods:

  1. A) The one-way ANOVA model was primarily used with the Fisher's PLSD to determine the nature of differences observed. Please include the ANOVA model p-values throughout the manuscript. They are missing.
  2. B) The repeated measures ANOVA is required for the analyses shown in Figure 3A, 3B, and 4C.
  3. C) Was a t-test used for 3D? If yes, please indicate this in the figure legend and add to the methods.

Response: Thank you very much for fundamental comments for statistics and I am sorry for incomplete description about statistics. I read the tutorial of statistical software, JMP and related textbooks and I decided to change the method from ANOVA and Fisher's PLSD to Tukey-Kramer’s HSD test that is thought to be a more sophisticated single-step multiple comparison procedure. I applied the test to Figures 2, 3, and 4 and changed the description in 2.7 Statistical analysis of Materials and Methods section. All p values were described in the legend whereas due to the limitation of the software, p value less than 0.0001 was expressed as p<0.0001.

6) Important background information on OTC deficiency was either missing or incorrect in the introduction and the discussion. I encourage the authors to carefully review more current literature to incorporate what the urea cycle is, inheritance patterns of OTCD, clinical presentations, long-term consequences, current treatments, in addition to other model organisms that have been reported for OTC deficiency. I made specific notes on this.

Response: Thank you very much for your suggestion and encouragement. We added what the urea cycle is, inheritance patterns of OTCD, clinical presentations, long-term consequences, current treatments, and mice model in Introduction and Discussion.

7) I made specific comments throughout a pdf version of the manuscript. Please review and revise per the comments.

Response: Thank you very much for your generous comments. We did our best to improve the manuscript according to your instructions.

8) Supplementary Table 1 - Revise title to "Pregnancy and Delivery Outcomes for OTC Carrier Female Pigs". Edit the font and sizes of the table content to be more consistent.

Response: According to your invaluable comment, I changed the title to ‘Pregnancy and Delivery Outcomes for OTC Carrier Female Pigs’.

Round 2

Reviewer 2 Report

Thanks for addressing my concerns. There are still a few items that need to be addressed.

1) The introduction is much improved. However, there are still a couple of issues.

1A) There appears to still be an incomplete sentence on line 51 (The mainstays of treatment are protein...). Please complete the sentence.

1B) Although some urea cycle disorders are associated with mitochondrial dysfunction in relevant tissues, they are not categorized as mitochondrial disorders. Mitochondrial disorders are a separate group of disorders from urea cycle disorders - this is well-known. Please fix the statements that refer to OTC deficiency and urea cycle disorders as mitochondrial disorders. 

2) I think you misunderstood my comments on the statistics. The overall approach used with the one-way ANOVA in the submitted manuscript was correct. When performing testing on multiple groups, one is to do the following: 1) Perform assumption testing to assess normality and equal variance. If the data is not skewed, you can perform the ANOVA using the raw data. If the data is skewed, perform a log transformation. 2) Perform the ANOVA. If significant main effects or a significant interaction is observed, perform the post-hoc testing. 3) Perform the post-hoc testing. 4) Report the p-values for the ANOVA model and the post-hoc testing. 

In the revised version, you only utilized post-hoc testing. This is not correct. Use of multiple comparisons with post-hoc testing increases the risk for type 1 error. This is the reason the ANOVA is used. Please use the ANOVA and post-hoc testing originally report. Please report the p-values for the ANOVA model and the post-hoc testing. This is important because one can only pursue post-hoc testing for multiple comparisons if the ANOVA model is significant.

As mentioned in my previous review, the ANOVA with repeated measures is needed for the analysis of the data in Figure 3A and 3 B, in addition to Figure 4C.

3) The discussion is much improved. Thank you for incorporating more literature. A couple more comments -- 

3A) Line 255 - Liver transplantation is considered a treatment for urea cycle disorders, not a cure. Despite liver transplantation, some individuals with urea cycle disorders cannot liberalize their diet, for example.  Additionally, liver transplantation may also not be permanent, due to graft rejection. Please revise this statement. 

3B) Lines 259-261- Please mention that safety, in addition to efficacy is still being assessed for cell and gene therapies.

3C) Line 266 - Should say pathogenic variant, not pathogen variant

3D) Lines 271-273 - This statement is not true. Please revise. Identification and diagnosis of OTC deficiency is still a challenge. Many people, particularly females with OTCD, may not diagnosed for many years. Sometimes, they do not receive a diagnosis until they are post-mortem due to sudden death related to hyperammonemia. See link for recent case example related to the difficulties diagnosing OTCD: https://www.sciencedirect.com/science/article/pii/S221442692030152X?via%3Dihub 

2E) Line 290: Dietary protein restriction and nitrogen scavenging agent therapy are the the mainstays of treatment for OTC deficiency.

2F) "..the advice of.."   - not advices

Author Response

Dear Reviewer 2
 Thank you very much for your careful review and crucial comments. We submit the revised manuscript according to your comments with track changes and the response to the comments in point-by-point fashion.

Thanks for addressing my concerns. There are still a few items that need to be addressed.
1) The introduction is much improved. However, there are still a couple of issues.
1A) There appears to still be an incomplete sentence on line 51 (The mainstays of treatment are protein...). Please complete the sentence.

I am very sorry to have submitted incomplete manuscript. I completed the sentence and we did our best to improve the contents according to your previous comment No.6. [Line 43 to 59 of revised version (7th of June, 2021 submitted)]

1B) Although some urea cycle disorders are associated with mitochondrial dysfunction in relevant tissues, they are not categorized as mitochondrial disorders. Mitochondrial disorders are a separate group of disorders from urea cycle disorders - this is well-known. Please fix the statements that refer to OTC deficiency and urea cycle disorders as mitochondrial disorders.

I am very sorry for my misunderstandings. I changed the phrases after I learned the definition of mitochondrial liver diseases. [Line 46 to 53]

2) I think you misunderstood my comments on the statistics. The overall approach used with the one-way ANOVA in the submitted manuscript was correct. When performing testing on multiple groups, one is to do the following: 1) Perform assumption testing to assess normality and equal variance. If the data is not skewed, you can perform the ANOVA using the raw data. If the data is skewed, perform a log transformation. 2) Perform the ANOVA. If significant main effects or a significant interaction is observed, perform the post-hoc testing. 3) Perform the post-hoc testing. 4) Report the p-values for the ANOVA model and the post-hoc testing. In the revised version, you only utilized post-hoc testing. This is not correct. Use of multiple comparisons with posthoc testing increases the risk for type 1 error. This is the reason the ANOVA is used. Please use the ANOVA and post-hoc testing originally report. Please report the p values for the ANOVA model and the post-hoc testing. This is important because one can only pursue post-hoc testing for multiple comparisons if the ANOVA model is significant. As mentioned in my previous review, the ANOVA with repeated measures is needed for the analysis of the data in Figure 3A and 3 B, in addition to Figure 4C.

Thank you very much for your precise instruction. I returned the statistical method to the original test, one-way ANOVA and Fisher’s protected least significant difference (PLSD) test. All p values of both ANOVA and post-hoc PLSD test were specified in figure legends. [Line 164 to 165, 2.7 Statistical analysis in Materials and methods, and legends of Figures 2, 3, and 4]

3) The discussion is much improved. Thank you for incorporating more literature. A couple more comments --
3A) Line 255 - Liver transplantation is considered a treatment for urea cycle disorders, not a cure. Despite liver transplantation, some individuals with urea cycle disorders cannot liberalize their diet, for example. Additionally, liver transplantation may also not be permanent, due to graft rejection. Please revise this statement.

I am sorry for a one-sided description. I improved the phrases to unbiased description. [Line 246 to 250]

3B) Lines 259-261- Please mention that safety, in addition to efficacy is still being assessed for cell and gene therapies.

Thank you very much for your comment. I added the word safety in the sentence. [Line 251]

3C) Line 266 - Should say pathogenic variant, not pathogen variant.

I am sorry for the word misuse. I changed ‘pathogen’ to ‘pathogenic’. [Line 257]

3D) Lines 271-273 - This statement is not true. Please revise. Identification and diagnosis of OTC deficiency is still a challenge. Many people, particularly females with OTCD, may not diagnosed for many years. Sometimes, they do not receive a diagnosis until they are post-mortem due to sudden death related to hyperammonemia. See link for recent case example related to the difficulties diagnosing OTCD: https://www.sciencedirect.com/science/article/pii/S221442692030152X?via%3Dihub

Thank you very much for your comment and hot information about intronic variant. I rephrased the statement by quoting the report you introduced. [Line 264 to 267]

2E) Line 290: Dietary protein restriction and nitrogen scavenging agent therapy are the mainstays of treatment for OTC deficiency.

I am sorry for incomplete description. I added ‘nitrogen scavenging agent treatment’ in the sentence. [Line 278 to 279]

2F) "..the advice of.." - not advices

I am sorry for the word error. I omitted ‘s’. [Line 281]
